# Psychometric Properties of Different Versions of the Body Shape Questionnaire in Female Aesthetic Patients

**DOI:** 10.3390/healthcare11182590

**Published:** 2023-09-20

**Authors:** Helder Miguel Fernandes, Patrícia Soler, Diogo Monteiro, Luís Cid, Jefferson Novaes

**Affiliations:** 1Polytechnic Institute of Guarda (IPG), 6300-559 Guarda, Portugal; 2Research Centre in Sports Sciences, Health Sciences and Human Development (CIDESD), 5001-801 Vila Real, Portugal; diogo.monteiro@ipleiria.pt (D.M.); luiscid@esdrm.ipsantarem.pt (L.C.); 3Department of Aesthetics and Cosmetology, Unigran Capital University, Campo Grande 79010-010, Brazil; patfisioterapia@hotmail.com; 4ESECS—Polytechnic of Leiria, 2411-901 Leiria, Portugal; 5Research Center in Quality of Life (CIEQV), 2001-904 Santarém, Portugal; 6Sport Science School of Rio Maior (ESDRM), Polytechnic of Santarém, 2040-413 Rio Maior, Portugal; 7Physical Education and Sports Department, Federal University of Rio de Janeiro (UFRJ), Rio de Janeiro 21941-901, Brazil; jeffsnovaes@gmail.com; 8Faculty of Physical Education and Sports, Federal University of Juiz de Fora, São Pedro 36036-900, Brazil

**Keywords:** body image, body shape concerns, factorial validity, internal consistency, female aesthetic patients

## Abstract

The purpose of this study was to examine and compare the psychometric properties (internal consistency and factorial validity) of different versions of the Body Shape Questionnaire (BSQ) in a sample of female aesthetic patients. The sample included 296 women attending aesthetic clinics, with ages ranging between 18 and 70 years (*M* = 32.23, *SD* = 11.35) and body mass index between 17.10 and 45.00 kg/m^2^ (*M* = 24.70, *SD* = 4.07). Nine different length versions of the BSQ (BSQ-34, BSQ-32, BSQ-16A, BSQ-16B, BSQ-14, BSQ-8A, BSQ-8B, BSQ-8C and BSQ-8D) were subjected to confirmatory factor analyses, using a robust maximum likelihood estimator. Robust fit indices indicated that the BSQ-8D version was the better-fitting and more parsimonious model (S-Bχ^2^/*df* = 1.81, CFI = 0.963, RMSEA = 0.052, SRMR = 0.043). This short version also showed appropriate reliability (McDonald’s omega and composite reliability = 0.87) and a very high correlation with the original BSQ-34 version (*r* = 0.95). In sum, these findings suggest that the BSQ-8D is the most valid, reliable and suitable BSQ version for measuring body shape concerns in female aesthetic patients.

## 1. Introduction

Over the last decades there has been substantial research and clinical interest in the psychological construct of body image. Body image is a multifaceted construct that comprises self-perceptions and attitudes of an individual regarding their own physical appearance, which may be described concisely as the “the psychological experience of embodiment” [1]. Although body image encompasses perceptual, affective, cognitive and behavioral aspects of body experience, most of the empirical research in this field has focused on the affective-evaluative dimension of body image [2], which includes concerns, appraisals and (dis)satisfaction with physical appearance, size and silhouette. Influenced by the cultural norms for appearance and unrealistic standards of beauty (i.e., thin or slim-thick ideal body types) [3], girls and women from western countries tend to report elevated rates of body image disturbance and physical appearance dissatisfaction [4,5], with a negative impact on their physical, social and mental health [6]. As such, there is a need for valid, reliable, specific and easy-to-apply questionnaires to assess body image concerns and dissatisfaction among women in both clinical and nonclinical settings.

One of the most widely used questionnaires that assesses body weight and shape concerns is the Body Shape Questionnaire (BSQ), which was developed by Cooper and colleagues [7]. The BSQ is a self-report measure that includes 34 items originally designed to capture the phenomenological experience of feeling fat, in persons with eating disorders or related body image problems [8]. This questionnaire includes questions that cover important body image symptoms, such as distressing preoccupation with weight and shape, embarrassment in public and avoidance of social activities or exposure of the body due to self-consciousness, excessive bodily feelings of fatness, as well as its antecedents and consequences [7].

The 34-item BSQ scale has been widely used and scored as a one-dimensional questionnaire, in clinical and nonclinical populations from different countries/languages [8,9,10,11,12]. The concurrent validity and reliability of several 34-item BSQ translations have been previously attested in several studies [2]. However, its factorial validity and length have been doubted, and subsequently several shorter versions of the original BSQ have been proposed. A study conducted by Evans and Dolan [13], after excluding the items 26 (self-induced vomiting) and 32 (use of laxatives), established two 16-item versions (i.e., BSQ-16A and BSQ-16B) and four 8-item versions of the questionnaire (i.e., BSQ-8A, BSQ-8B, BSQ-8C and BSQ-8D). These derived versions showed appropriate psychometric properties and very high correlations with the BSQ-34 total scores, ranging from 0.96 to 0.99. The psychometric properties of these short versions have been reported in several studies, with some concluding that two of the 8-items versions (BSQ-8B and BSQ-8D) performed the best [14,15], whereas others indicated that the BSQ-8C version showed higher sensitivity to change during therapy [16]. In addition, Downson and Henderson [17] investigated the construct validity of a 14-item version of the BSQ designed by one of the original authors of the scale (i.e., P.J. Cooper). This short version showed good reliability and convergent validity in a small clinical sample. Other studies also provided psychometric support for the BSQ-14 in Swedish [18] and Norwegian [19] clinical and nonclinical samples. In view of this lack of a clear consensus, more research is needed on the psychometric properties of the BSQ original and derived versions, especially in specific at-risk groups of individuals susceptible to excessive concerns and dissatisfaction with body image.

Body image dissatisfaction is a significant public health issue that is associated with pursuing cosmetic and/or aesthetic enhancing procedures with the purpose of physical beautification [6,12,20]. Body weight and shape concerns are associated with appearance-management behaviors that may include physical activity, pharmacotherapy and diet, as well as body contouring procedures, such as liposuction, abdominoplasty, breast augmentation and rhinoplasty [21]. Recent results of the International Society of Aesthetic Plastic Surgery (ISAPS) annual report [22] indicated that the USA and Brazil are the two top countries in surgical and nonsurgical cosmetic procedures, accounting for 33.0% of the total aesthetic procedures performed worldwide in 2021, with women representing 86.5% of the total procedures. In the specific case of Brazil, the same report indicated a total of 1,634,220 surgical procedures and 1,089,420 nonsurgical procedures, with botulinum toxin, hyaluronic acid, liposuction, breast augmentation, eyelid surgery and abdominoplasty being the most performed aesthetic procedures.

Understanding the body shape and weight concerns and disturbances of individuals undergoing aesthetic procedures is a core aspect of the clinical care of these patients and their families [23]. As such, there is a need for valid and reliable psychometric measures of body image dissatisfaction in this specific field [20,24], allowing for a better assessment and knowledge of the patients’ psychological functioning before, during and after the treatment. Moreover, this assessment should be part of a comprehensive screening that may also help identify unrealistic preoperative motivations and postoperative expectations, as well as potential associated psychiatric disorders (e.g., body dysmorphic disorder and eating disorders), all of which can contraindicate the aesthetic treatment [20,23]. Previous studies have provided empirical support for the use of shorter versions of the original BSQ [10,13,14,16,17,25], making it a more practical, convenient and easy-to-administer measure of body shape dissatisfaction in diverse settings. However, no study has yet investigated the psychometric properties of the different versions of the BSQ, specifically in women seeking aesthetic or cosmetic procedures, with most of the studies including clinical samples (with anorexia nervosa or bulimia nervosa) [16,17,19], university students [11,14,15,26] or the general population [18,25]. Therefore, the main aim of this study is to investigate and compare the psychometric properties (i.e., internal consistency and factorial validity) of the different versions of the BSQ in a sample of female aesthetic patients.

## 2. Materials and Methods

### 2.1. Participants

A sample of female patients attending aesthetic clinics located in Campo Grande, Mato Grosso do Sul, Brazil, was randomly recruited to investigate the psychometric properties of the original BSQ and derived versions. Sample size was estimated based on the recommendations of 5 to 10 respondents per parameter to be estimated and a minimum of 200 individuals for psychometric testing purposes [27,28].

A total of 296 women, with a mean age of 32.23 years (*SD* = 11.35; range from 18 to 70) and a mean body mass index (BMI) of 24.70 kg/m^2^ (*SD* = 4.07; range from 17.10 to 45.00) volunteered to participate in this study. Most women reported being single (47.3%) or married (43.6%), and fewer indicated being divorced (7.4%) or widowed (1.7%). Regarding the educational level, 6.4% of the women had up to 9 years of formal schooling, 27.0% had up to 12 years, 42.2% had incomplete graduation and the remaining 24.4% had completed graduation. Of the respondents, nearly three-quarters of the sample (74.0%) were employed and 26.0% were unemployed. Concerning the socioeconomic status, 58.8% reported a monthly income of less than 2500 Reais (1 Brazilian Real equals about 0.20 US Dollar), 30.7% indicated a monthly income between 2500 and 5000 Reais, whereas the remaining 10.5% received a monthly income above 5000 Reais. Regarding the type of cosmetic procedures received, a higher number of women reported nonsurgical procedures (66.6%), such as nonsurgical skin tightening (12.8%), nonsurgical fat reduction (10.5%), chemical peel (9.8%), cellulite treatment (7.8%), hyaluronic acid (4.4%), and the combination of two or more treatments (21.3%). Conversely, 60.5% of the total sample also indicated that they had already undergone surgical cosmetic procedures, namely: breast augmentation (15.9%), abdominoplasty (11.5%), liposuction (10.8%), rhinoplasty (5.7%), eyelid surgery (5.4%), buttock augmentation (4.4%) and two or more surgical procedures (6.8%).

The inclusion criteria of the study were the following: (a) sex (female); (b) 18 years of age or older; (c) current attendance to aesthetic or cosmetic clinics in Campo Grande, Brazil; and (d) no current or past diagnosis of psychiatric disorder and/or neurological disease.

This study was conducted according to the Declaration of Helsinki and its revisions, and followed the ethical regulations established by Resolution 466/2012 of the National Health Council. The research project was approved by the National Council of Ethics in Research (CONEP), CAAE: 21844213.7.0000.0021.

### 2.2. Procedures

After receiving approval from the research ethics committee, the City Hall of Campo Grande, Mato Grosso do Sul, Brazil, was approached with the aim of identifying all inpatient and outpatient aesthetic clinics located in the city. Based on this survey, a total of twelve clinics were randomly sampled and their managing directors were contacted, providing authorization for data collection.

Women attending these selected aesthetic clinics were approached and invited to participate in the study. Willing participants, who fulfilled the inclusion criteria, received information related to the purpose of the research (influencing factors of body image in women) and data collection procedures. Next, they were invited to complete a composite of questionnaires, including sociodemographic information and the 34-item BSQ, in a quiet room.

Participation in the study was voluntary, anonymous and without compensation. Signed written consent was obtained from all recruited women before participation.

### 2.3. Measures

Self-report data were initially collected for sociodemographic characteristics (age, marital status, education level, monthly income, employment status and use/type of aesthetic procedures), current or past diagnosis of psychiatric disorder and/or neurological disease and for anthropometric measurements (weight and height). The BMI was computed as weight (in kilograms) divided by height (in meters) squared.

Participants also completed the Brazilian version [29] of the Body Shape Questionnaire [7]. The BSQ includes 34 items measuring cognitive, affective and behavioral antecedents and consequences of body size and shape concerns during the last four weeks. Examples of some items are: Item 2, “Have you been so worried about your shape that you have been feeling you ought to diet?”; Item 24, “Have you worried about other people seeing rolls of fat around your waist or stomach?”; and Item 31, “Have you avoided situations where people could see your body (e.g., communal changing rooms or swimming baths)?”. Responses are rated using a six-point Likert scale ranging from 1 (never) to 6 (always). The item scores of the different versions (original and derived) are summed to obtain the total scores, which may range from 34 to 204 (BSQ-34), from 16 to 96 (BSQ-16A and BSQ-16B), from 14 to 84 (BSQ-14) and from 8 to 48 (BSQ-8A, BSQ-8B, BSQ-8C and BSQ-8D). Previous studies have demonstrated moderate to good psychometric properties of the BSQ in Brazilian adolescents [30] and university students [15,29].

### 2.4. Models’ Description

In this study, nine measurement BSQ models were analyzed and compared. The list of items included in each model is presented in Table 1.

### 2.5. Statistical Analyses

A preliminary inspection of the data was conducted for accuracy and missing values. The missing value analysis indicated 0.14% of missing data, only related to some of the BSQ items, which were imputed (i.e., replaced) with the median of each item. Next, descriptive statistics (mean, median, mode and standard deviation) and univariate (skewness and kurtosis) and multivariate normality (Mardia’s normalized multivariate kurtosis) were computed. Absolute values of univariate skewness <|2|, univariate kurtosis <|7| and Mardia’s normalized multivariate kurtosis <|3| were considered indicative of normality [28,31].

A series of confirmatory factor analyses (CFA) were conducted, using the EQS 6.4 software (Multivariate Software, CA, USA), in order to assess and compare the factorial validity of the measurement models. Given the non-normal distribution of categorical data, a robust estimation method (robust Maximum Likelihood with Satorra–Bentler scaled χ^2^) was used [31,32]. The assessment of the models’ fit was based on multiple criteria [28,33] and included the following robust fit indices: Satorra–Bentler scaling correction chi-square/degrees of freedom (S-Bχ^2^/*df*), Comparative Fit Index (CFI), Root Mean Square Error of Approximation (RMSEA) and its 90% confidence interval (CI), and the Akaike Information Criterion (AIC). The Standardized Root Mean Square Residual (SRMR) was also considered. Taking into consideration the sample size (*N* = 296), model complexity (one factor with 8 to 34 items) and degrees of error in model specification (20 to 527), the following cut-off values were considered as evidence of relatively good fit to the data [33] (p. 642): S-Bχ^2^/*df* < 3, CFI > 0.94, RMSEA < 0.07 and SRMR < 0.08. A lower AIC value indicates a better model fit.

McDonald’s omega (ω) and composite reliability (CR) coefficients were calculated to evaluate the internal consistency of the different BSQ versions, with values > 0.70 being considered appropriate. Pearson’s correlation analyses were used to assess the association between the original and derived (shortened) versions of the BSQ.

Unless otherwise mentioned, statistical analyses were performed using IBM SPSS 27.0 (IBM Corp, Armonk, NY, USA) and statistical significance was set at *p* < 0.05.

## 3. Results

Table 2 presents the descriptive statistics for all items of the BSQ.

Descriptive results showed that the most endorsed items by the sample were numbers 34 (“ought to exercise”), 4 (“afraid of becoming fat(ter)”) and 5 (“flesh being not firm enough”). On the other hand, the less endorsed items were numbers 26 (“vomited to feel thinner”), 27 (“taking up too much room in the company of others”) and 18 (“not going out to social occasions”). Univariate normality results indicated that nine items (7, 8, 10, 13, 18, 25, 26, 27 and 32) had inadequate indices of skewness and/or kurtosis.

Table 3 presents the results of the CFAs conducted for the nine measurement models of the BSQ, indicated in Table 1.

Results of the CFAs indicated that only two models simultaneously fulfilled the multiple criteria requirements for good fit, with model BSQ-8D showing a relatively better fit (higher CFI and lower RMSEA, SRMR and AIC) compared with model BSQ-8B. The longer versions (34 and 32 items) and the shortened 16- and 14-items versions of the BSQ had smaller CFI (i.e., <0.92) and higher AIC values, showing a poorer fit to the data. Conversely, BSQ-8A and BSQ-8C models showed the worst RMSEA and 90% CI results with values ranging up to 0.119 and 0.104, respectively.

The standardized factor loadings of the better-fit model (i.e., BSQ-8D) ranged between 0.54 (item 18) and 0.78 (item 14).

Table 4 presents the Pearson’s correlation matrix and internal consistency (McDonald’s omega) coefficients for all tested versions of the BSQ.

Internal consistency results showed high McDonald’s omega and composite reliability coefficients (ω and CR > 0.80), indicating appropriate reliability for all versions of the BSQ. All derived versions of the BSQ had very high correlations with the original 34-item version, with Pearson’s coefficients ranging between 0.95 and 1.00. The better-fitting model (i.e., version BSQ-8D) showed a very high coefficient of determination with the original BSQ-34 version (*R*^2^ = 90%).

## 4. Discussion

The main aim of the present study was to evaluate the reliability and factorial validity of different short and full-length versions of the BSQ in female aesthetic patients. To the best of our knowledge, this is the first study attempting to investigate and compare the psychometric properties of different versions of this questionnaire in women attending aesthetic or cosmetic clinics and, consequently, identify the best version to be used by researchers and/or clinicians in this population.

One of the main findings of this study is that the BSQ-8D version demonstrated more favorable psychometric properties, in terms of factorial validity, when compared to the original and other derived versions of the BSQ scale. Previous studies have also provided support for the BSQ-8D [15,34], whereas others have favored the BSQ-8B [14] or the BSQ-8C version [10,16]. When considering the item composition of the BSQ-8D version, it is noteworthy that most of the items focus on general body dissatisfaction, body shame related to dieting and body dissatisfaction associated with appearance-related social comparison and social avoidance. These items represent important preoccupations and concerns with body shape and/or weight (i.e., a negative appearance evaluation) that helps explain why women pursue body contouring procedures [20,35]. Moreover, the current results have not provided support for the use of longer versions (i.e., BSQ-34, BSQ-32, BSQ-16A, BSQ-16B and BSQ-14) in female aesthetic patients, in line with previous studies that question the usefulness of the complete 34-item BSQ [13,14,16]. The administration of a shorter, more valid and reliable body shape questionnaire version in aesthetic clinical contexts appears to be more useful as part of a more detailed and thorough psychosocial assessment, in an initial consultation and/or a preoperative intervention that should also include other areas, such as the patients’ motivations and expectations, psychiatric status and other clinical information [36,37]. Such a comprehensive psychological assessment of patients before aesthetic or cosmetic procedures plays a significant role in planning and following-up the intervention [12,20].

Another main finding of this study was that the shortened version BSQ-8D also showed high internal consistency (ω and CR = 0.87) and a very high correlation with the original BSQ-34 version (*r* = 0.95), which suggests that reducing the number of items did not compromise reliability or narrow the body dissatisfaction construct represented by the BSQ. These findings are in accordance with results obtained from previous studies [15,16] which also question the usefulness of administrating the complete 34-item BSQ in certain settings [13,14,16]. Possible explanations for these findings include the fact that the 34-item version may have unnecessary items with similar characteristics that estimate body shape concerns [15] or the fact that the content of items not included in the BSQ-8D represent less valued aspects of body shape dissatisfaction in the present sample, therefore resulting in a more favorable factor structure. Nevertheless, more research is needed to investigate other validity and reliability evidence for the BSQ-8D in the aesthetic population.

The current study has some strengths and limitations to be acknowledged. This study is the first to identify the BSQ short version that presents better psychometric properties in female aesthetic patients, which represents the largest group of persons undergoing surgical and nonsurgical cosmetic procedures [22,35]. This finding may contribute to monitoring the treatment progress and related changes in women’s body shape dissatisfaction over time. One other strength of the study was the use of a diverse sample in terms of age, BMI and type of surgical and nonsurgical cosmetic procedures received. On the other hand, the study limitations include the cross-sectional design of this study, not allowing for the test–retest analysis of the instrument. Furthermore, this study did not include men. However, a notable increase in male aesthetic treatments has been documented in recent years [38]. Therefore, future studies should also aim to investigate the psychometric properties of the different versions of the BSQ in men and assess the measurement invariance of the selected BSQ version by sex.

## 5. Conclusions

From the results of the present study, it can be concluded that the shortened BSQ-8D version showed better factorial validity in a sample of female aesthetic patients than the remaining original and derived versions. Moreover, this 8-item version of the BSQ also showed a high internal consistency and convergence with the original BSQ-34 version, providing additional support for its usefulness in aesthetic clinical settings.

Researchers and clinicians are encouraged to use the brief, valid and reliable BSQ-8D version to assess, monitor and evaluate antecedents and consequences of body image concerns in women seeking or undergoing aesthetic/cosmetic surgical and nonsurgical intervention, as part of a more comprehensive patient record.

## Figures and Tables

**Table 1 healthcare-11-02590-t001:** Measurement models assessed in the study.

Abbreviation	Items
BSQ-34 [7]	1–34
BSQ-32 [13]	1–25, 27–31, 33, 34
BSQ-16A [13]	1, 3, 5, 7, 8, 9, 10, 11, 15, 17, 20, 21, 22, 25, 28, 34
BSQ-16B [13]	2, 4, 6, 12, 13, 14, 16, 18, 19, 23, 24, 27, 29, 30, 31, 33
BSQ-14 [17]	2, 9, 12, 14, 17, 19, 20, 21, 23, 24, 25, 29, 31, 34
BSQ-8A [13]	1, 3, 7, 8, 9, 10, 17, 34
BSQ-8B [13]	5, 11, 15, 20, 21, 22, 25, 28
BSQ-8C [13]	4, 6, 13, 16, 19, 23, 29, 33
BSQ-8D [13]	2, 12, 14, 18, 24, 27, 30, 31
BSQ-34 [7]	1–34
BSQ-32 [13]	1–25, 27–31, 33, 34

**Table 2 healthcare-11-02590-t002:** Descriptive statistics of the BSQs’ items.

Items	Mean	SD	Median	Mode	Skewness	Kurtosis
BSQ1	2.62	1.28	3.00	3.00	0.79	0.57
BSQ2	2.98	1.58	3.00	3.00	0.53	−0.60
BSQ3	2.47	1.57	2.00	1.00	0.83	−0.29
BSQ4	3.38	1.66	3.00	3.00	0.20	−1.00
BSQ5	3.02	1.56	3.00	3.00	0.49	−0.64
BSQ6	2.64	1.63	2.00	1.00	0.77	−0.46
BSQ7	1.61	1.12	1.00	1.00	2.26	5.24
BSQ8	1.47	1.09	1.00	1.00	2.66	6.75
BSQ9	1.76	1.26	1.00	1.00	1.77	2.59
BSQ10	1.70	1.35	1.00	1.00	2.08	3.39
BSQ11	1.75	1.20	1.00	1.00	1.78	2.80
BSQ12	2.16	1.39	2.00	1.00	1.16	0.69
BSQ13	1.50	1.00	1.00	1.00	2.49	6.50
BSQ14	2.34	1.48	2.00	1.00	1.05	0.30
BSQ15	2.93	1.56	3.00	3.00	0.61	−0.51
BSQ16	2.33	1.65	2.00	1.00	1.08	0.06
BSQ17	2.52	1.61	2.00	1.00	0.89	−0.23
BSQ18	1.43	1.05	1.00	1.00	2.81	7.71
BSQ19	1.87	1.43	1.00	1.00	1.76	2.21
BSQ20	2.28	1.38	2.00	1.00	1.13	0.82
BSQ21	2.67	1.63	2.00	1.00	0.73	−0.51
BSQ22	2.22	1.64	1.00	1.00	1.19	0.17
BSQ23	2.60	1.74	2.00	1.00	0.79	−0.67
BSQ24	2.90	1.69	3.00	2.00	0.67	−0.71
BSQ25	1.62	1.25	1.00	1.00	2.23	4.39
BSQ26	1.20	0.68	1.00	1.00	4.31	21.53
BSQ27	1.40	1.05	1.00	1.00	2.99	8.72
BSQ28	2.94	1.68	3.00	3.00	0.54	−0.79
BSQ29	2.30	1.42	2.00	1.00	1.06	0.47
BSQ30	2.44	1.48	2.00	1.00	0.90	0.05
BSQ31	2.45	1.62	2.00	1.00	1.01	−0.05
BSQ32	1.54	1.17	1.00	1.00	2.38	5.11
BSQ33	2.24	1.38	2.00	1.00	1.17	0.84
BSQ34	3.42	1.69	3.00	3.00	0.25	−1.05

**Table 3 healthcare-11-02590-t003:** Fit indices for the BSQ measurement models.

Model	S-Bχ^2^	*df*	S-Bχ^2^/*df*	CFI	RMSEA (90% CI)	SRMR	AIC
BSQ-34	1102.92	527	2.09	0.816	0.061 (0.056–0.066)	0.063	48.92
BSQ-32	1017.08	464	2.19	0.823	0.064 (0.058–0.069)	0.061	89.08
BSQ-16A	246.15	104	2.37	0.880	0.068 (0.057–0.079)	0.059	38.15
BSQ-16B	238.86	104	2.30	0.901	0.066 (0.055–0.077)	0.056	30.86
BSQ-14	209.53	77	2.72	0.913	0.076 (0.064–0.089)	0.051	55.53
BSQ-8A	74.37	20	3.72	0.851	0.096 (0.073–0.119)	0.066	34.37
BSQ-8B	41.84	20	2.09	0.960	0.061 (0.034–0.087)	0.043	1.84
BSQ-8C	57.91	20	2.90	0.939	0.080 (0.056–0.104)	0.049	17.91
BSQ-8D	36.19	20	1.81	0.963	0.052 (0.023–0.079)	0.043	−3.81

**Table 4 healthcare-11-02590-t004:** Pearson’s correlation and internal consistency coefficients.

BSQ Version	ω	CR	BSQ-34	BSQ-32	BSQ-16A	BSQ-16B	BSQ-14	BSQ-8A	BSQ-8B	BSQ-8C	BSQ-8D
BSQ-34	0.97	0.96	–	1.00	0.99	0.99	0.98	0.95	0.95	0.97	0.95
BSQ-32	0.97	0.96		–	0.99	0.99	0.98	0.95	0.96	0.97	0.95
BSQ-16A	0.92	0.92			–	0.95	0.95	0.96	0.97	0.93	0.91
BSQ-16B	0.94	0.94				–	0.97	0.91	0.92	0.98	0.97
BSQ-14	0.94	0.94					–	0.92	0.92	0.95	0.95
BSQ-8A	0.84	0.85						–	0.86	0.89	0.87
BSQ-8B	0.87	0.87							–	0.91	0.88
BSQ-8C	0.89	0.89								–	0.89
BSQ-8D	0.87	0.87									–

Note: All correlations significant at *p* < 0.001. ω = McDonald’s omega; CR = composite reliability.

## Data Availability

Data is available upon request from the corresponding author.

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
