# Peer review of "Psychometric Properties of Different Versions of the Body Shape Questionnaire in Female Aesthetic Patients"

_healthcare, 2023, doi:10.3390/healthcare11182590_

Round 1

Reviewer 1 Report

Dear Authors,

I have a few minor suggestions for improvement of your manuscript.

First, you could more clearly defend the importance of the study.

You could explain a bit more about recruitment - what were people told about the purpose of the study? How was it determined that people had no psychiatric diagnosis?

It would be helpful to give examples of BSQ items in the method section.

It would be helpful to know know more about the body satisfaction in this group - perhaps descriptive statistics on the eight items for the recommended scale?

Some awkward wording:

line 97 - women "attending" aesthetic or cosmetic settings - a different word than attending?

Line 200 and 202 - "valued" items - endorsed might be a better word

Author Response

  1. Summary

Thank you very much for taking the time to review this manuscript. We have carefully read and addressed all your comments. Please find the detailed responses below and the corresponding revisions/corrections highlighted/in track changes in the re-submitted file.

  1. Point-by-point response to Comments and Suggestions for Authors

Comments 1: Dear Authors, I have a few minor suggestions for improvement of your manuscript. First, you could more clearly defend the importance of the study.

Response 1: Thank you for pointing out this issue. We agree with this comment and, therefore, have included the following text in the final paragraph of the Introduction section: “Moreover, this assessment should be part of a comprehensive screening that may also help identify unrealistic preoperative motivations and postoperative expectations, as well as potential psychiatric disorders associated (e.g., body dysmorphic disorder and eating disorders), all of which can contraindicate the aesthetic treatment [20,23].”. (lines 93 to 96)

Comments 2: You could explain a bit more about recruitment - what were people told about the purpose of the study? How was it determined that people had no psychiatric diagnosis?

Response 2: We thank you for highlighting these issues. More detailed information on the sample’s recruitment procedures and data collection was provided in the 2.2 and 2.3 sections, respectively. (lines 151 to 152 and 160 to 161)

Comments 3: It would be helpful to give examples of BSQ items in the method section.

Response 3: Examples of three BSQ items were included in the 2.3 section. (lines 166 to 170)

Comments 4: It would be helpful to know more about the body satisfaction in this group - perhaps descriptive statistics on the eight items for the recommended scale?

Response 4: Descriptive statistics on the eight items for the recommended scale are presented in Table 2.

  1. Response to Comments on the Quality of English Language

Some awkward wording:

line 97 - women "attending" aesthetic or cosmetic settings - a different word than attending?

Line 200 and 202 - "valued" items - endorsed might be a better word

Response: Thank you for pointing out these issues. Both suggestions were considered in the revised version of the manuscript. (lines 101, 213 and 215)

Reviewer 2 Report

Overall this is a well thought out and well presented piece of work on an area worth investigating. Anyway we can reduce scale redundancy is a good thing!

Well written, an appropriate method (with a couple of reservations) and the analysis was conducted very well. No issues there

My couple of concerns relate to the "value" of the paper. Yep, you've re-tested the BSQ in another setting (women undergoing aesthetic procedures). And you've found that the BSQ-8D has the best validity/reliability and fit to data - for your sample. But as you say other studies support and not support the BSQ-8D as the most appropriate form of the scale - probably due to context if we unpack those studies. So the conclusion it should be used in populations of women seeking aesthetic cosmetic procedures is appropriate.

However, as you note, this is a very specific sample (Brazilian as well, rather than cross-cultural across national cultures).

My main question relates to HOW is the administration of the BSQ-8D of value to women undertaking such procedures? Is this merely an "academic" exercise? Will it shift end user perceptions? Or will it simply identify that women with body image issues are more likely to seek aesthetic procedures? How will knowing that the BSQ-8D is the most reliable measure of the BSQ help inform our understanding of body image issues for women undertaking aesthetic procedures? Then how will THAT help to mitigate those issues - if possible? 

I think a bit more exploration of that notion is warranted. Your limitations are pretty sparse currently, so I think there is space there for this question to be pondered.

Author Response

  1. Summary

Thank you very much for taking the time to review this manuscript. We have carefully read and addressed all your comments. Please find the detailed responses below and the corresponding revisions/corrections highlighted/in track changes in the re-submitted file.

  1. Point-by-point response to Comments and Suggestions for Authors

Comments 1: Overall this is a well thought out and well presented piece of work on an area worth investigating. Any way we can reduce scale redundancy is a good thing!

Well written, an appropriate method (with a couple of reservations) and the analysis was conducted very well. No issues there.

My couple of concerns relate to the "value" of the paper. Yep, you've re-tested the BSQ in another setting (women undergoing aesthetic procedures). And you've found that the BSQ-8D has the best validity/reliability and fit to data - for your sample. But as you say other studies support and not support the BSQ-8D as the most appropriate form of the scale - probably due to context if we unpack those studies. So the conclusion it should be used in populations of women seeking aesthetic cosmetic procedures is appropriate.

However, as you note, this is a very specific sample (Brazilian as well, rather than cross-cultural across national cultures).

My main question relates to HOW is the administration of the BSQ-8D of value to women undertaking such procedures? Is this merely an "academic" exercise? Will it shift end user perceptions? Or will it simply identify that women with body image issues are more likely to seek aesthetic procedures? How will knowing that the BSQ-8D is the most reliable measure of the BSQ help inform our understanding of body image issues for women undertaking aesthetic procedures? Then how will THAT help to mitigate those issues - if possible?

I think a bit more exploration of that notion is warranted. Your limitations are pretty sparse currently, so I think there is space there for this question to be pondered.

Response 1: Thank you for your positive feedback. We agree with your comments and concerns. Therefore, we have included the following text in the final paragraph of the Introduction section to highlight the importance of using a short version of the BSQ as part of initial and subsequent consultations: “Moreover, this assessment should be part of a comprehensive screening that may also help identify unrealistic preoperative motivations and postoperative expectations, as well as potential psychiatric disorders associated (e.g., body dysmorphic disorder and eating disorders), all of which can contraindicate the aesthetic treatment [20,23].” (lines 93 to 96). Therefore, we consider that the use of a reduced version of the BSQ, alongside other measures, may contribute to a more comprehensive assessment of the women’s psychological functioning throughout all phases of the treatment. In addition, we have also reformulated the Limitations topic, as suggested, to provide a better understanding of the relevance of using a reduced version of the BSQ in women undergoing aesthetic surgical and/or non-surgical procedures. (lines 282 to 287)

Reviewer 3 Report

Many thanks for submitting your manuscript for review. Overall, I found this paper to be excellently written throughout. The argument is very tight and clear, and the rationale develops well.

In the participants section, I wonder if it would aid clarity to put the demographic information in a table rather than in text? There is a little bit of trepetition in this section and I think the inclusion criteria and sample size estimates would go better at the beginning, then refer to the demographic indormation. 

The results section is thorough and clear. I wonder if the paragraph on the discussion of the descriptive data and the associated table would be better placed before the paragraph discussing the approach to the inferential analyses, i.e. from line 176 on page 4. I found myself going back to the paragraph discussing the descriptive data (lines 168-175 on page 4) when reading lines 200-205 on page 6.

I find it very interesting that you mention the lack of male participants as a limitation of the current study. I agree with this statement but I also think this is a strength of your research - I would perhaps expect to see different items being more vs less important in a male sample, and this may therefore have confounded your results. I think this would be a great avenue for further researh - though difficult to recruit participants. I worry a little that most readers would think this point is a little shallow and perhaps you could clarify the relevance of it and why it is actually an important point to note here. 

Author Response

  1. Summary

Thank you very much for taking the time to review this manuscript. We have carefully read and addressed all your comments. Please find the detailed responses below and the corresponding revisions/corrections highlighted/in track changes in the re-submitted file.

  1. Point-by-point response to Comments and Suggestions for Authors

Comments 1: Many thanks for submitting your manuscript for review. Overall, I found this paper to be excellently written throughout. The argument is very tight and clear, and the rationale develops well.

In the participants section, I wonder if it would aid clarity to put the demographic information in a table rather than in text? There is a little bit of repetition in this section and I think the inclusion criteria and sample size estimates would go better at the beginning, then refer to the demographic information.

Response 1: In line with your comment, we have now included the sample size estimates before the demographic information (lines 111 to 113). As the sample’s demographic information is now comprised in a single paragraph, we consider no need to present this information in an additional table.

Comments 2: The results section is thorough and clear. I wonder if the paragraph on the discussion of the descriptive data and the associated table would be better placed before the paragraph discussing the approach to the inferential analyses, i.e. from line 176 on page 4. I found myself going back to the paragraph discussing the descriptive data (lines 168-175 on page 4) when reading lines 200-205 on page 6.

Response 2: We understand the concern of the reviewer. However, the order of presentation is in line with other studies analyzing the BSQ’s psychometric properties, so we choose to maintain the original order.

Comments 3: I find it very interesting that you mention the lack of male participants as a limitation of the current study. I agree with this statement, but I also think this is a strength of your research - I would perhaps expect to see different items being more vs less important in a male sample, and this may therefore have confounded your results. I think this would be a great avenue for further research - though difficult to recruit participants. I worry a little that most readers would think this point is a little shallow and perhaps you could clarify the relevance of it and why it is actually an important point to note here.

Response 3: Thank you for pointing this out. We agree with this comment. Therefore, we have reformulated the Limitations topic, as suggested, to provide a better understanding of the relevance of using a reduced version of the BSQ in women undergoing aesthetic surgical and/or non-surgical procedures, as well as the need for future studies examining only a male sample. (lines 282 to 293)